# A New 1′ × 1′ Global Seafloor Topography Model Predicted from Satellite Altimetric Vertical Gravity Gradient Anomaly and Ship Soundings BAT_VGG2021

**Minzhang Hu [1,2,3,\*], Li Li [1,3], Taoyong Jin [4], Weiping Jiang [4], Hanjiang Wen [5] and Jiancheng Li [4]**

1   Key Laboratory of Earthquake Geodesy, Institute of Seismology, CEA, Wuhan 430071, China; ynulili@foxmail.com
2   School of Earth Sciences, Institute of Disaster Prevention, Langfang 065201, China
3   Wuhan Gravitation and Solid Earth Tides, National Observation and Research Station, Wuhan 430071, China
4   School of Geodesy and Geomatics, Wuhan University, Wuhan 430071, China; tyjin@sgg.whu.edu.cn (T.J.); wpjiang@sgg.whu.edu.cn (W.J.); jcli@sgg.whu.edu.cn (J.L.)
5   Chinese Academy of Surveying & Mapping, Beijing 100001, China; wenhanjiang@163.com
\*   Correspondence: mzhhu@whu.edu.cn

**Abstract:** In this paper, we construct a new 1′ × 1′ global seafloor topography model, BAT_VGG2021, using the satellite altimetric vertical gravity gradient anomaly model (VGG), SIO curv_30.1.nc, and ship soundings. Approximately 74.66 million single-beam depths and more than 180 GB of multibeam grids were downloaded and adopted from the National Centers for Environmental Information (NCEI), Japan Agency for Marine-Earth Science and Technology (JAMSTEC), and Geosciences Australia (GA). The SIO curv_30.1.nc model was used to predict seafloor topography at 15~160 km wavelengths, and ship soundings were used to calibrate topography to VGG ratios. The accuracy of the new BAT_VGG2021 model was assessed by comparing it with ship soundings and existing models. The results indicate that the standard deviation of differences between the predicted model and ship soundings is about 40~80 m, and ~93% of the differences are within 100 m, similar to that of the SIO topo_20.1.nc model. The new BAT_VGG2021 model shows better accuracy than the DTU18BAT, ETOPO1, and GEBCO_08 models, and has been improved significantly from our last model, BAT_VGG2014.

**Keywords:** seafloor topography; satellite altimetry; vertical gravity gradient anomaly; ship soundings

## 1. Introduction

The seafloor covers ~71% of the solid earth and has diverse tectonic features. Knowledge of the seafloor topography (seawater depth) plays a pivotal role in geosciences research, resource exploitation, and environmental protection, etc. However, mapping the seafloor requires a significant investment of labor and money.

Traditionally, seafloor topography can be measured directly by echo sounder systems equipped on a ship. However, it is time-consuming work to measure the global seafloor due to the slow speed of the ship. Scientists have suggested that people have surveyed only ~18% of the seafloor, at an effective resolution of ~1 km, in the past centuries [1]. In the latest SRTM+V2.1 seafloor topography model at 15 arc-second resolution, only ~10.84% of the seafloor is directly constrained by acoustic surveys [2]. It is assessed that hundreds of ship-years and billions in terms of financial cost are required to map the global seafloor using echo sounding systems [1,3]. Thus, it is very difficult to map the oceans at a proper resolution directly by ship currently.

Fortunately, the development of satellite altimetry technology provides an indirect way to recover seafloor features at a moderate resolution and accuracy. Altimeters measure the height of the ocean's surface, which can be used to derive gravity anomaly. Then the gravity anomaly can be used to predict seafloor topography at short-to-middle

wavelengths [4–6]. Since the launching of the Skylab in 1973, many satellite altimetric missions, such as Geosat, ERS, and Jason, have been carried out successfully. The altimetric derived gravity models have been updated constantly with the accumulation of satellite altimetry data and improvements in the data processing technology [7–12]. At the same time, a series of global seafloor topography models, constructed with altimetric data and ship soundings, were published by the Scripps Institute of Oceanography (SIO), Danmarks Tekniske Universitet (DTU), Wuhan University (WHU), and the International Hydrographic Organization (IHO), etc. [2,4–6,13–16].

At present, most of the publicly available seafloor topography models, such as SIO topo_20.1.nc and DTU18BAT, have been predicted from altimetric gravity anomalies. While scientists have suggested that the vertical gravity gradient anomaly (VGG) may be used to strengthen seafloor topography at short wavelengths [17], few papers have been published [18–21]. For more than ten years, we have engaged in seafloor topography model construction using ship soundings and VGG and published a global model, BAT_VGG2014, in 2014 [22]. Presently, new altimetry data from AltiKa, CryoSat, and Sentinel-3A/B have been collected to derive a new version of the VGG model, SIO curv_30.1.nc. In this paper, we construct a new $1' \times 1'$ global seafloor topography model using ship soundings and the latest version of the VGG model. Approximately 74.66 million single-beam depths and ~6.6 GB of multibeam grids were downloaded from the National Centers for Environmental Information (NCEI), ~120 GB of multibeam grids were downloaded from the Japan Agency for Marine-earth Science and Technology (JAMSTEC), and ~54 GB of multibeam grids were downloaded from Geosciences Australia (GA). The accuracy of the new model was assessed by comparing it with ship soundings and existing models such as SIO topo_20.1.nc, DTU18BAT, ETOPO1, GEBCO_08, and BAT_VGG2014 [23,24].

## 2. Theory and Methods

According to the lithospheric flexural isostasy theory [25], seamounts loading will introduce flexure of the oceanic crust (Moho discontinuity)

$$R(k) = -H(k)\frac{(\rho_c - \rho_w)}{(\rho_m - \rho_c)}\Phi'_e(k) \tag{1}$$

where $R(k)$ and $H(k)$ represent the Fourier transform of Moho flexure and seafloor undulations; $\rho_m$, $\rho_c$, $\rho_w$ are densities of the mantle, crust, and water, respectively; $k = 2\pi/\lambda$ is the wavenumber, and $\lambda$ is the wavelength; $\Phi_e(k)$ is the lithospheric flexural response function [26]

$$\Phi_e(k) = \left[\frac{Dk^4}{(\rho_m - \rho_c)g} + 1\right]^{-1} \tag{2}$$

where $D$ is lithospheric flexural rigidity, $D = ET_e^3/[12(1 - v^2)]$, and $E$ is Young's modulus, $T_e$ is lithospheric effective elastic thickness; $v$ is Poisson's ratio; $g$ is the average gravity acceleration.

Seamounts and the corresponding Moho flexure introduce most of the gravity anomalies on the sea surface. Based on Parker's formula [27], these gravity anomalies, $\Delta G(k)$, can be given by

$$\Delta G(k) = 2\pi G(\rho_c - \rho_w)e^{-kd}\sum_{n=1}^{\infty}H^n(k) + 2\pi G(\rho_m - \rho_c)e^{-k(t+d)}\sum_{n=1}^{\infty}R^n(k) \tag{3}$$

where $G$ is gravitation constant, $d$ is mean water depth, and $t$ is mean crustal thickness. This series expansion formula converges very quickly. Substituting Equation (1) into Equation (3), and considering $n = 1$, Equation (3) can be simplified as

$$\Delta G(k) = 2\pi G(\rho_c - \rho_w)e^{-kd}\left(1 - \Phi_e(k)e^{-kt}\right)\cdot H(k) \tag{4}$$



In the frequency domain, the VGG, $\Delta G_z(k)$, will be

$$\Delta G_z(k) = 2\pi G(\rho_c - \rho_w)e^{-kd}k\left(1 - \Phi_e(k)e^{-kt}\right)\cdot H(k) \tag{5}$$

Thus, the transform functions between seafloor and gravity anomaly, $Z_{topo-grav}(k)$, or VGG, $Z_{topo-grad}(k)$, are

$$\begin{cases} Z_{topo-grav}(k) = 2\pi G(\rho_c - \rho_w)e^{-kd}\left(1 - \Phi_e(k)e^{-kt}\right) \\ Z_{topo-grad}(k) = 2\pi G(\rho_c - \rho_w)e^{-kd}k\left(1 - \Phi_e(k)e^{-kt}\right) \end{cases} \tag{6}$$

These transform functions are composed by coefficient, $2\pi G(\rho_c - \rho_w)$, exponential decay function, $exp(k)$, isostatic response function, $\varphi(k)$, and wavenumber, $k$, where

$$\begin{cases} exp(k) = e^{-kd} \\ \varphi(k) = 1 - \Phi_e(k)e^{-kt} \\ \tau(k) = k \end{cases} \tag{7}$$

Using parameters in Table 1, the exponential decay function, $exp(k)$, works like a low-pass filter (thick blue line in Figure 1), and suggests high-frequency topography signal is suppressed due to upward-continuation of seawater depth. The isostatic response function, $\varphi(k)$, works as a high-pass filter (red lines in Figure 1), and indicates the gravity signal is weakened by isostatic compensation mass in the oceanic crust. The combined effect of these functions results in a band-pass filter of transform function, as shown in Figure 2. Figure 2 indicates that compared to $Z_{topo-grav}(k)$, the transform function between seafloor topography and VGG, $Z_{topo-g\,rad}(k)$, suppresses the effect of isostasy and enlargesthe signal at shorter wavelengths (<~100 km).

**Table 1.** Theoretical crustal model for calculating admittance $Z_{topo-grav}(k)$ and $Z_{topo-grad}(k)$.

| Parameters | Notation | Value |
|---|---|---|
| Density of water | $\rho_w$ | 1030 kg/m$^3$ |
| Density of crust | $\rho_c$ | 2800 kg/m$^3$ |
| Density of mantle | $\rho_m$ | 3350 kg/m$^3$ |
| Mean crustal thickness | $T$ | 7 km |
| Mean water depth | $D$ | 4 km |
| Effective elastic thickness | $T_e$ | 3, 5, 10, 25 km |
| Young's modulus | $E$ | $10^{11}$N/m$^2$ |
| Poisson's ratio | $v$ | 0.25 |

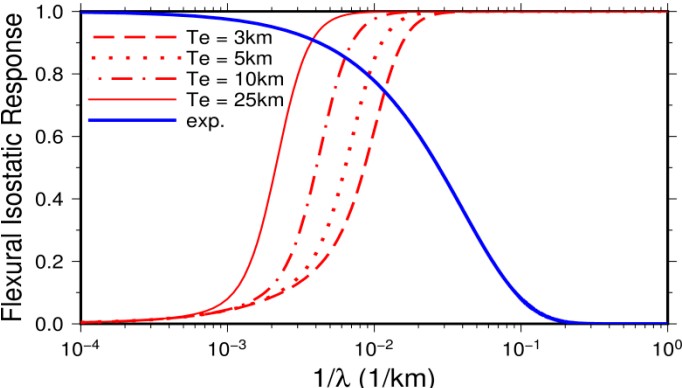

**Figure 1.** The red lines represent response functions, $\varphi(k)$, with the lithospheric effective elastic thickness of 3 km, 5 km, 10 km, and 25 km, respectively, which work like high-pass filters. The thick blue line shows the exponential decay function, $exp(k)$, which works like a low-pass filter.

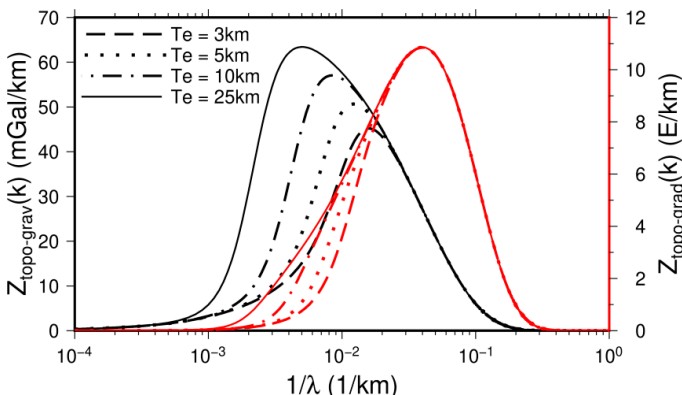

**Figure 2.** The black lines represent transform functions between seafloor topography and gravity anomalies, $Z_{topo-grav}(k)$, with lithospheric effective elastic thickness of 0 km, 3 km, 5 km, 10 km, and 25 km, respectively. The red lines show transform functions between seafloor topography and VGG, $Z_{topo-grad}(k)$. The results show that the transform functions work like band-pass filters. The transform function, $Z_{topo-grad}(k)$, suppresses the effect of isostasy and enlarge signal at wavelengths shorter than ~100 km.

Isostasy analysis of seafloor features suggested that the seafloor topography shows high coherence with gravity anomalies at certain short-to-middle wavelength bands [28,29]. For example, in the northwestern Pacific (144°~180°E, 0°~36°N), Figure 3 indicates that the seafloor topography and gravity anomaly or VGG show high coherence at certain wavelength bands. Thus, the satellite altimetric gravity anomalies or VGG were used to constrain seafloor topography at 10~160 km or 20~200 km wavelength bands [4,5,19].

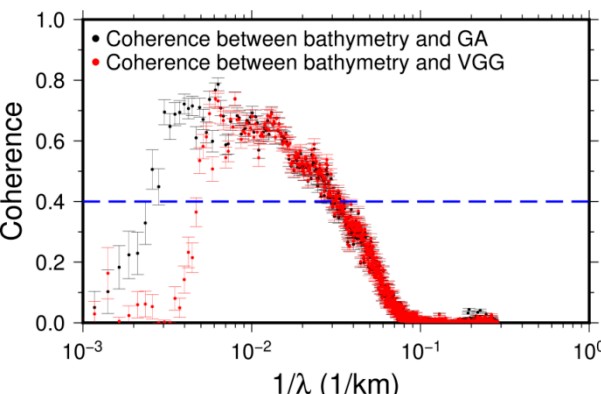

**Figure 3.** The coherence between seafloor topography and gravity anomaly (black dots) or VGG (red dots) in the northwestern Pacific (144°~180°E, 0°~36°N). This indicates that the seafloor topography and gravity anomaly or VGG show high coherence at certain wavelength bands. At long wavelengths (>500 km), the topography–VGG coherence is lower than the topography–GA coherence.

A band-pass filter was designed to process gravity anomaly or VGG, and seafloor topography at certain wavelength bands can be predicted by

$$
\begin{cases}
H(k) = \frac{1}{2\pi G(\rho_c-\rho_w)} \cdot e^{kd} \cdot BF(\Delta G(k)) \\
H(k) = \frac{1}{2\pi G(\rho_c-\rho_w)} \cdot \frac{e^{kd}}{k} \cdot BF(\Delta G_z(k))
\end{cases}
\tag{8}
$$

where *BF* indicates a band-pass filter.

## 3. Data and Results

### 3.1. Data Sources

In this paper, the VGG model, existing seafloor topography model, and ship soundings were used to construct a new global seafloor topography model. The latest version of the SIO VGG model, curv_30.1.nc, which includes new altimetric data from AltiKa, CryoSat, and Sentinel-3A/B satellites, was used to constrain seafloor topography at 15~160 km wavelength bands, as shown in Figure 4. The SIO topography model, topo_20.1.nc, was used to control seafloor topography at wavelengths longer than 160 km. The ship soundings were used to calibrate topography-to-VGG ratios at 15~160 km wavelength bands.

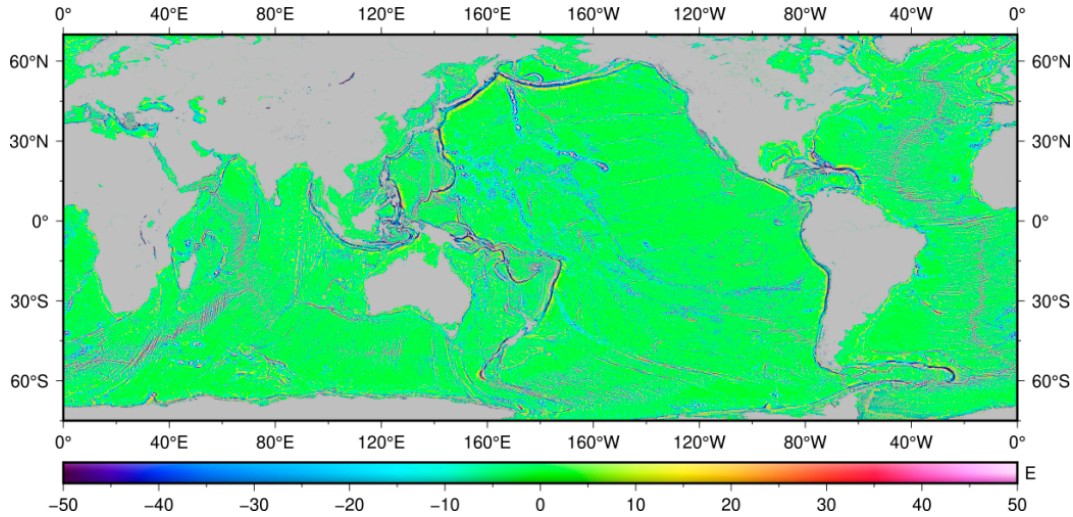

**Figure 4.** The SIO VGG model, curv_30.1.nc, from ftp://topex.ucsd.edu/pub/global_grav_1min/, released on 9 October 2020. This version includes an additional year of AltiKa, CryoSat, and Sentinel-3A/B data than the previous version.

The ship soundings, including single-beam points and multibeam grids, were downloaded from NCEI, JAMSTEC, and GA, as show in Figure 5. The NCEI provided ~74.66 million single-beam points (black dots) in more than 3800 cruise surveys collected since 1960Sand ~6.6 GB of multibeam grids (blue dots). Approximately 120 GB of multibeam grids were downloaded from the JAMSTEC (yellow dots), with most of the data distributed around Japan and the northwestern Pacific. Multibeam grids, ~54 GB, around Australia were obtained from GA (purple dots). These ship depths distribute unevenly over the world, and most of them are in the northern hemisphere. All data sources are listed in Table 2.

The ship soundings show an uneven distribution of data quality all over the world oceans, mostly due to the use of analog echo sounders and poor positing before the availability of satellite navigation [14,30]. Smith (1993) assessed the accuracy of ship soundings collected between 1955 and 1992 in Lamont–Doherty Earth Observatory and found the least accurate data were in the southern oceans where the median of the crossover errors at intersecting ship tracks were 100–250 m [30]. The modern multibeam grids from the NCEI, JAMSTEC, and GA have a relative accuracy of about 1% [1]. The single-beam depths were cleaned and edited by scientists at the NCEI before being provided to the public, but there are still a few significant errors. However, we focused on constructing a global seafloor topography model, rather than editing and correcting ship soundings. Thus, we cleaned the ship soundings and removed ship data with obvious errors by comparing them with the SIO topo_20.1.nc model, before constructing the new seafloor topography model. We calculated the standard deviation (STD) of differences between the ship soundings and the SIO topo_20.1.nc model. In each $2° \times 2°$ segment, ship depths with ship-model differences larger than twice of the STD were deleted. In the global oceans ($-180°$~$180°$E, $-75°$~$70°$N), ~5% of the single-beam points were deleted.The multibeam grids were re-sampled to cover ~160 million 15 arc-second grids using the GMT blockmedian [31]. Approximately 90% of

the cleaned ship points were applied to construct the new model, and 10% were used to assess the model accuracy.

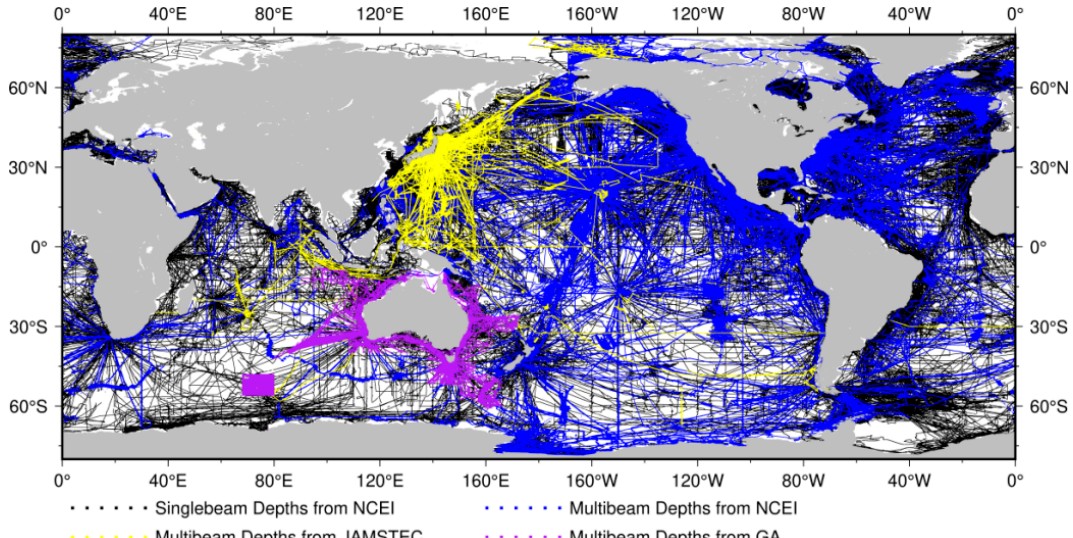

**Figure 5.** Global distribution of the ship soundings. The black dots indicate single-beam depths from NCEI, the blue dots indicate multibeam depths from NCEI, the yellow dots indicate multibeam depths from JAMSTEC, and the purple dots indicate multibeam depths from GA.

**Table 2.** Data sources used in this paper.

| Data Sources | Descriptions | Processes | Data Provider |
|---|---|---|---|
| Topography modelSIO topo_20.1.nc | Seafloor topography at 1 arc-minute resolution derived from altimetric gravity anomalies. | Low-pass filtered to construct model at wavelengths longer than 160 km. | SIO, UCSD |
| Altimetric VGGSIO curv_30.1.nc | Model derived from satellite altimetric missions at 1 arc-minute resolution. | Band-pass filtered and downward continued to constrain seafloor topography at 15–160 km wavelength bands | |
| Multibeam grids | Shipboard multibeam grid. | Re-sampled at each 15 arc-second grid | JAMSTEC |
| Multibeam grids | AusSeabed-2018 at 50 m resolution; MH370 searching seafloor topography at 150 m resolution; Kerguelen seafloor topography model at 100 m resolution; Macquarie seafloor topography model at 100 m resolution. | Re-sampled at each 15 arc-second grid | GA |
| Multibeam grids | Depth grids at about 100 m ~ 2 km resolution, provided by website AutoGrid service. | Re-sampled at each 15 arc-second grid | NCEI |
| Single-beam depths | ~74.66 million points | Evaluated by comparing with SIO topo_20.1.nc model | NCEI |

### 3.2. Data Processing Procedure

The following processing steps were applied to manipulate the data and construct a new $1' \times 1'$ global seafloor topography model, as shown in Figure 6.

1. The SIO topo_20.1.nc model was filtered to construct a reference model at wavelengths longer than 160 km, $h_{long}(x)$. Then, the reference depths at the ship points, $h_{ref\_ship}(x')$, were interpolated from $h_{long}(x)$.

2. At ship points, the residual depths, $h_{resi\_ship}(x')$, can be calculated by subtracting $h_{ref\_ship}(x')$ from the observed depths, $h_{ship}(x')$.

$$h_{resi_{ship}}(x') = h_{ship}(x') - h_{ref\_ship}(x') \tag{9}$$

3. The SIO curv_30.1.nc model was band-pass filtered, downward continued, and divided by $k$ to construct VGG at 15~160 km wavelength bands, $\Delta G_{z\_down}(x)$, and then was used to interpolate VGG at the ship points, $\Delta G_{z\_ship}(x')$.

4. The topography-to-VGG ratios at the ship points were calculated by

$$s(x') = h_{resi\_ship}(x') / \Delta G_{z\_ship}(x') \tag{10}$$

The ratios were then gridded to a $1' \times 1'$ grid, $S(x)$.

5. The gridded ratios, $S(x)$, and band-pass filtered VGG, $\Delta G_{z\_down}(x)$, were used to constrain seafloor topography at 15~160 km wavelength bands,

$$h_{pre}(x) = S(x) \cdot \Delta G_{z\_down}(x) \tag{11}$$

6. The final seafloor topography model becomes

$$h(x) = h_{long}(x) + h_{pre}(x) \tag{12}$$

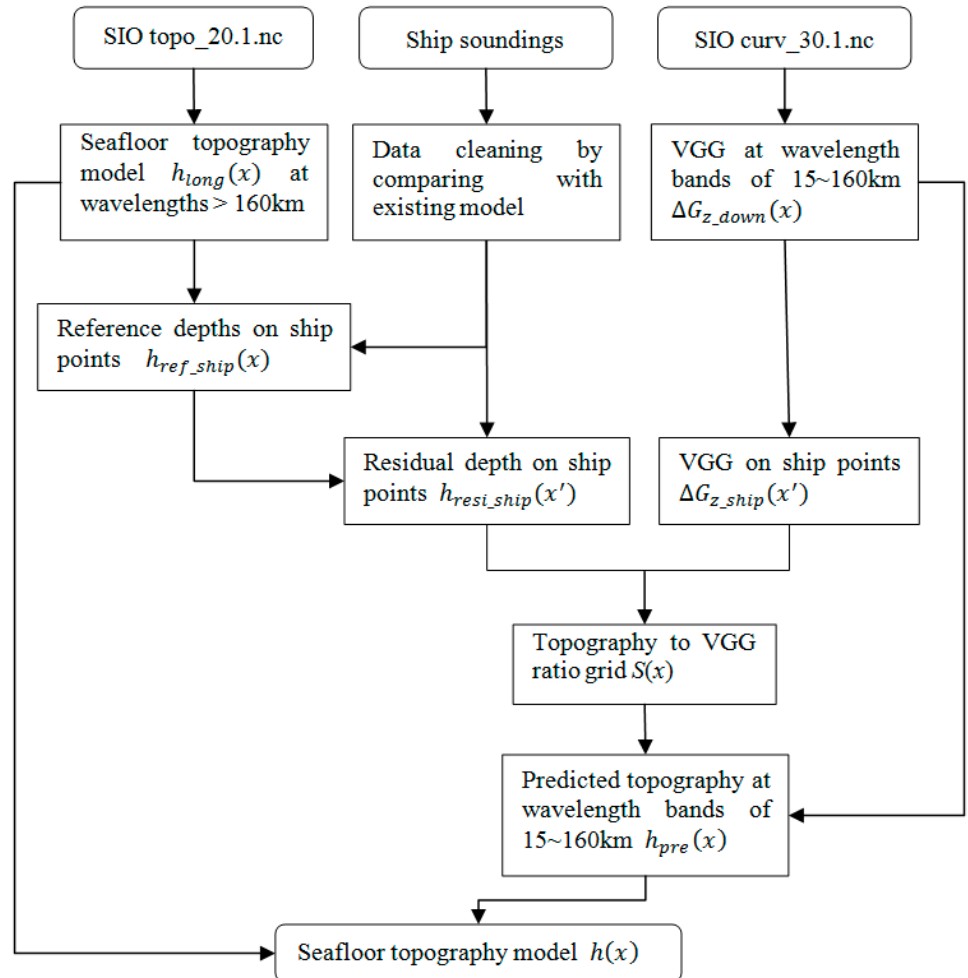

**Figure 6.** Data processing flow chart of seafloor topography construction from ship soundings and the altimetric VGG. The latest version of SIO model, topo_20.1.nc, was used to constrain seafloor

topography model at wavelengths longer than 160 km. The ship soundings were cleaned by comparing with topo_20.1.nc. About 90% of the cleaned ship points were applied to constrain topography to VGG ratios at 15~160 km wavelength bands, and 10% were used to assess the model accuracy. The VGG model, curv_30.1.nc was used to predict seafloor topography model at 15~160 km wavelength bands.

For example, Figure 7 shows seafloor topography prediction results in the northwestern Pacific (144°~180°E, 0°~36°N). The seamounts, such as Marcus-Wake Guyots, were recovered very well. Compared with ship depths, the differences between the predicted model and ship depths have a mean of −0.182 m and a standard deviation of 35.561 m. That means the data processing procedure was properly performed and VGG can be used to construct a seafloor topography model with high accuracy.

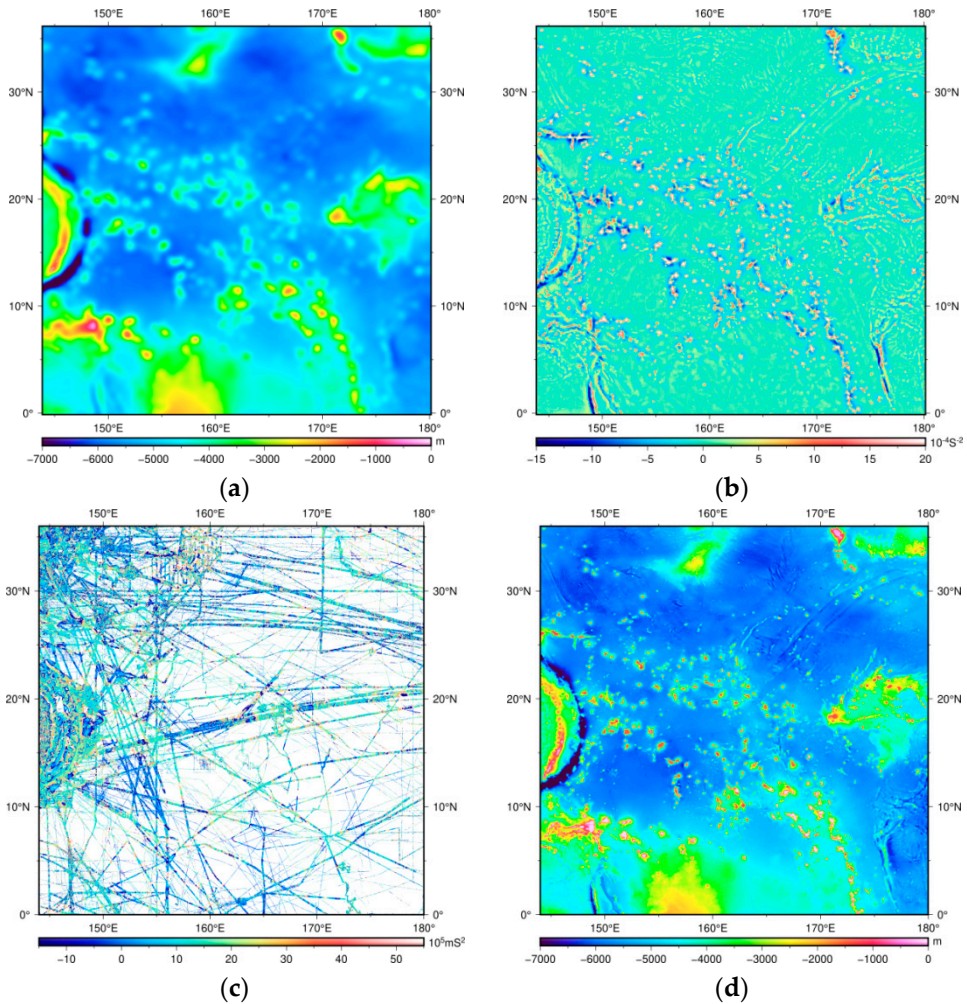

**Figure 7.** Seafloor topography construction example in the northwestern Pacific (144°~180°E, 0°~36°N). The seafloor topography at wavelengths longer than 160 km, $h_{long}(x)$ (**a**), the VGG grid at 15~160 km wavelength bands, $\Delta G_{z\_down}(x)$ (**b**), the topography-to-VGG ratios at ship points $s(x')$ (**c**), and the predicted $1' \times 1'$ seafloor topography model, $h(x)$ (**d**).

### 3.3. The New $1' \times 1'$ Global Seafloor Topography Model

The new $1' \times 1'$ global seafloor topography model, BAT_VGG2021, was predicted from VGG and ship soundings, as shown in Figure 8. The model clearly revealed seafloor features, such as mid-ocean ridges, seafloor plateaus, and seamount chains. Figure 9 shows the global distribution of differences between BAT_VGG2021 and ship soundings. The results indicate that the new predicted model fits ship measurements very well. The standard deviation of model–ship differences is 45.464 m, and ~93% of the differences

are within 100 m. Figure 10 shows the differences between BAT_VGG2021 and the SIO topo_20.1.nc model. The standard deviation difference of these two models is 80.732 m, ~84% of the differences are within 100 m, and ~95.8% of the differences are within 200 m. The frequency distribution histograms of the differences between BAT_VGG2021 and ship soundings, and the differences between BAT_VGG2021 and the SIO topo_20.1.nc model, are shown in Figure 11. These results indicate that VGG can be used to constrain seafloor topography at 15~160 km wavelength bands, and the data processing method proposed in this paper is correct.

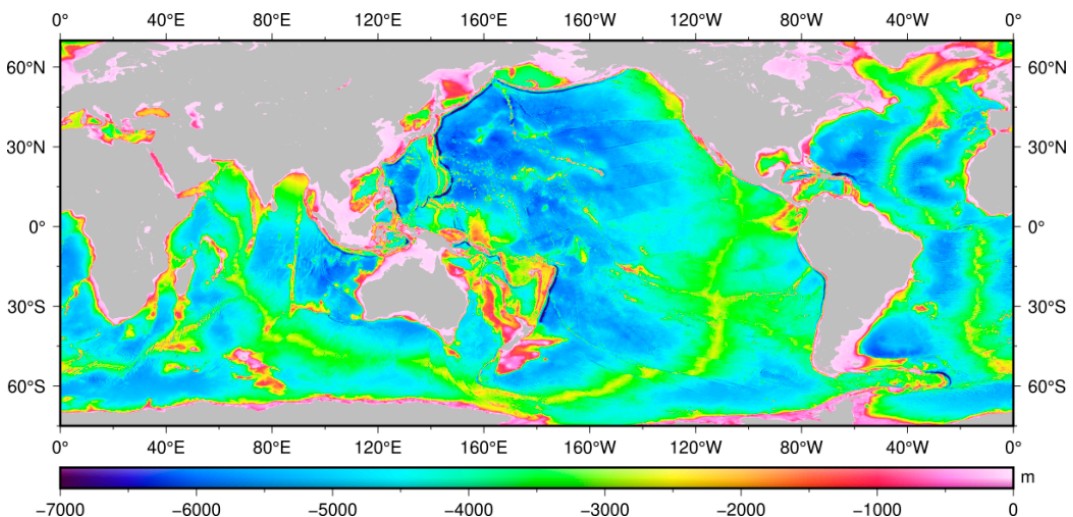

**Figure 8.** The new $1' \times 1'$ global seafloor topography model predicted from VGG and ship soundings, BAT_VGG2021.

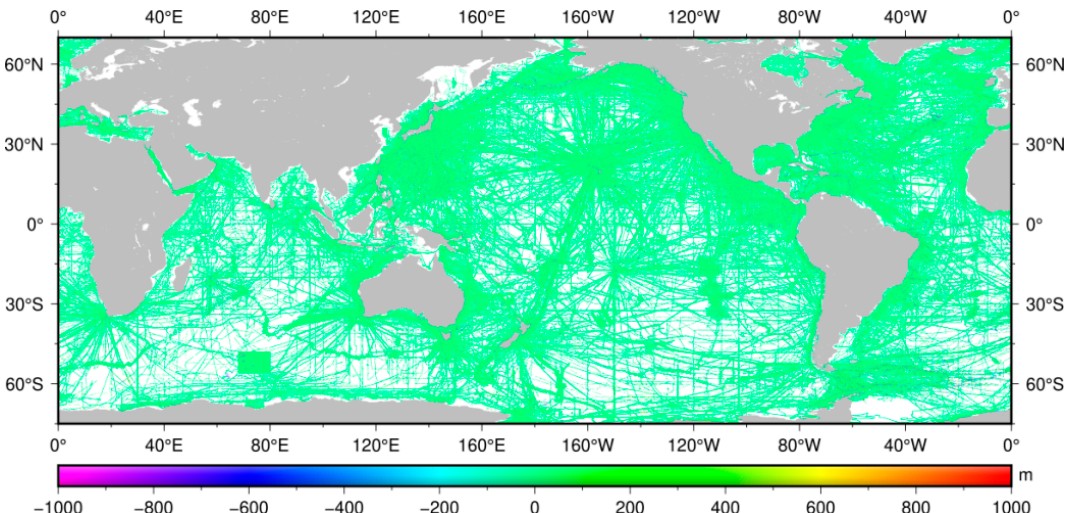

**Figure 9.** Global distribution of the differences between BAT_VGG2021 and ship soundings. The standard deviation of model–ship differences is 45.464 m, and ~93% of the differences are within 100 m. The result indicated that the predicted model fits ship measurements very well.

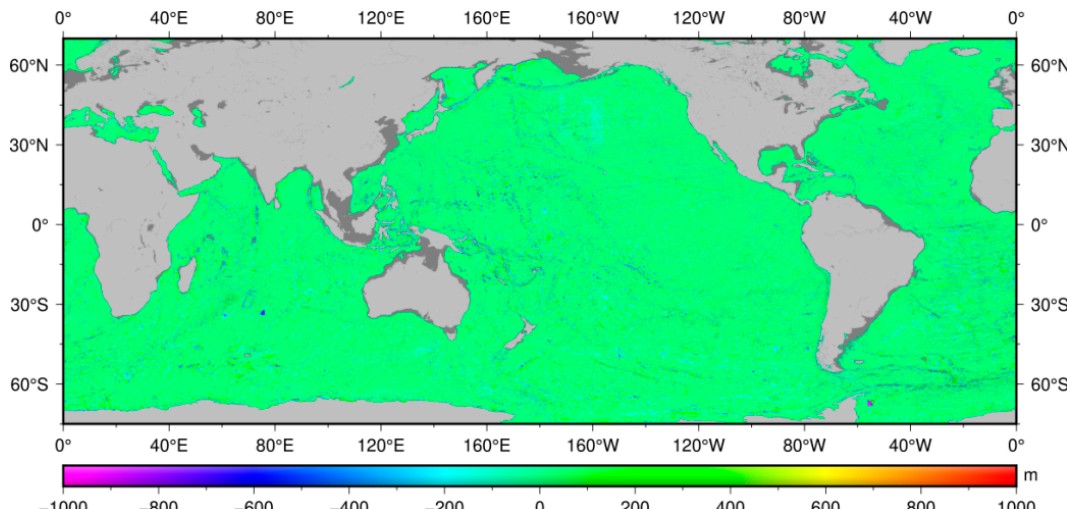

**Figure 10.** The differences between BAT_VGG2021 and the SIO topo_20.1.nc model. The standard deviation difference of these two models is 80.732 m, ~84% of the differences are within 100 m, and ~95.8% of the differences are within 200 m.

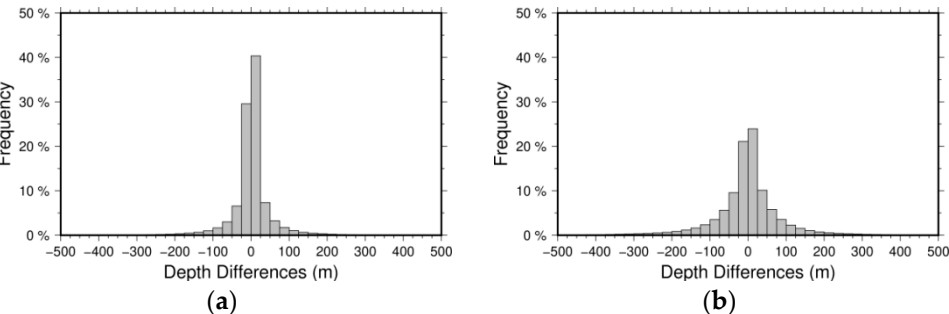

**Figure 11.** The frequency distribution histogram of the differences between BAT_VGG2021 and ship soundings (**a**), and differences between BAT_VGG2021 and SIO topo_20.1.nc model (**b**).

## 4. Discussion

### 4.1. Accuracy Evaluated by Comparing with Ship Soundings and Existing Models

We can assess the accuracy of the BAT_VGG2021 model by comparing it with ship soundings and existing models, such as SIO topo_20.1.nc, DTU18BAT, ETOPO1, GEBCO_08, and our last model BAT_VGG2014. The topo_20.1.nc model is the latest version of the global seafloor topography model released by SIO and is being predicted from ship soundings and satellite altimetric gravity anomalies. DTU18BAT is the latest version of the seafloor topography model released by DTU and is being constructed by a combination of the GEBCO model and satellite altimetric gravity anomalies. ETOPO1 is a global topography model released by National Geophysical Data Center, and an old version of the SIO seafloor topography model was used to fill depths in world oceans [24]. GEBCO_08 model is gridded from digital contours. BAT_VGG2014 is a model predicted from single-beam soundings and the satellite altimetric VGG model [22].

We compared existing models with ship soundings in the north Pacific (120°~280°E, 0°~70°N), south Pacific (120°~300°E, −75°~0°N), north Atlantic (280°~360°E, 0°~70°N), south Atlantic (−60°~20°E, −75°~0°N), and Indian Ocean (20°~120°E, −75°~26°N), respectively. Table 3 summarizesthe differences between existing global seafloor topography models and ship soundings in these five regions. The results show that the STDs of differences between the new BAT_VGG2021 model and ship soundings are about 40~80 m, with 39.6 m in the north Pacific and 77.6 m in the south Atlantic. The northern hemisphere shows better accuracy than in the southern hemisphere, mainly due to there being more ship depths covered in the north. The results suggest that the BAT_VGG2021

model has accuracy similar to the SIO topo_20.1.nc model and significantly better than the ETOPO1 and GEBCO_08 models. The accuracy has been improved obviously from our last BAT_VGG2014 model, which has STD differences of about 120~160 m, due to more altimetric data and multibeam depths being used. Figure 12 shows the frequency distribution of the differences between BAT_VGG2021 and ship soundings in the five regions. It indicates that more than 93% of the differences are within 100 m, except for ~87.74% in the south Atlantic.

**Table 3.** The statistics of differences between global seafloor topography models and ship soundings.

| Region | Model | Minimums (m) | Maximums (m) | Mean (m) | STD (m) |
|---|---|---|---|---|---|
| North Pacific (120°~280°E, 0°~70°N) | BAT_VGG2021 | −204.6 | 204.5 | 0.7 | 39.6 |
| | SIO topo.20.1.nc | −207.3 | 207.3 | −0.2 | 45.0 |
| | DTU18BAT | −315.1 | 315.1 | 5.7 | 65.8 |
| | BAT_VGG2014 | −507.7 | 507.7 | 22.2 | 125.1 |
| | ETOPO1 | −497.1 | 497.1 | 11.2 | 119.7 |
| | GEBCO_08 | −757.9 | 757.9 | 29.6 | 184.7 |
| South Pacific (120°~300°E, −75°~0°N) | BAT_VGG2021 | −246.0 | 246.0 | 1.1 | 46.7 |
| | SIO topo.20.1.nc | −258.7 | 258.7 | 1.3 | 53.1 |
| | DTU18BAT | −420.6 | 420.6 | 8.0 | 91.2 |
| | BAT_VGG2014 | −595.0 | 595.1 | 30.6 | 156.2 |
| | ETOPO1 | −612.2 | 612.2 | 11.9 | 159.5 |
| | GEBCO_08 | −926.8 | 926.8 | 20.2 | 225.9 |
| North Atlantic (280°~360°E, 0°~70°N) | BAT_VGG2021 | −201.6 | 201.7 | 1.3 | 39.8 |
| | SIO topo.20.1.nc | −219.1 | 219.1 | 0.1 | 49.2 |
| | DTU18BAT | −288.0 | 288.0 | 3.1 | 63.6 |
| | BAT_VGG2014 | −451.3 | 451.3 | 14.1 | 119.0 |
| | ETOPO1 | −480.5 | 480.6 | 5.2 | 116.2 |
| | GEBCO_08 | −642.4 | 642.0 | 28.0 | 161.7 |
| South Atlantic (−60°~20°E, −75°~0°N) | BAT_VGG2021 | −508.0 | 508.0 | 0.8 | 77.6 |
| | SIO topo.20.1.nc | −319.2 | 319.3 | 1.8 | 54.5 |
| | DTU18BAT | −400.5 | 400.5 | 4.3 | 71.4 |
| | BAT_VGG2014 | −519.3 | 519.0 | 10.2 | 120.1 |
| | ETOPO1 | −546.1 | 546.1 | 6.7 | 126.1 |
| | GEBCO_08 | −753.0 | 753.0 | 24.9 | 192.9 |
| Indian Ocean (20°~120°E, −75°~26°N) | BAT_VGG2021 | −232.6 | 232.6 | 1.3 | 41.7 |
| | SIO topo.20.1.nc | −264.1 | 264.2 | −0.2 | 55.6 |
| | DTU18BAT | −420.7 | 420.7 | 6.0 | 100.1 |
| | BAT_VGG2014 | −593.9 | 593.9 | 20.9 | 160.3 |
| | ETOPO1 | −581.9 | 581.9 | 7.6 | 150.5 |
| | GEBCO_08 | −663.1 | 663.1 | 15.7 | 166.3 |

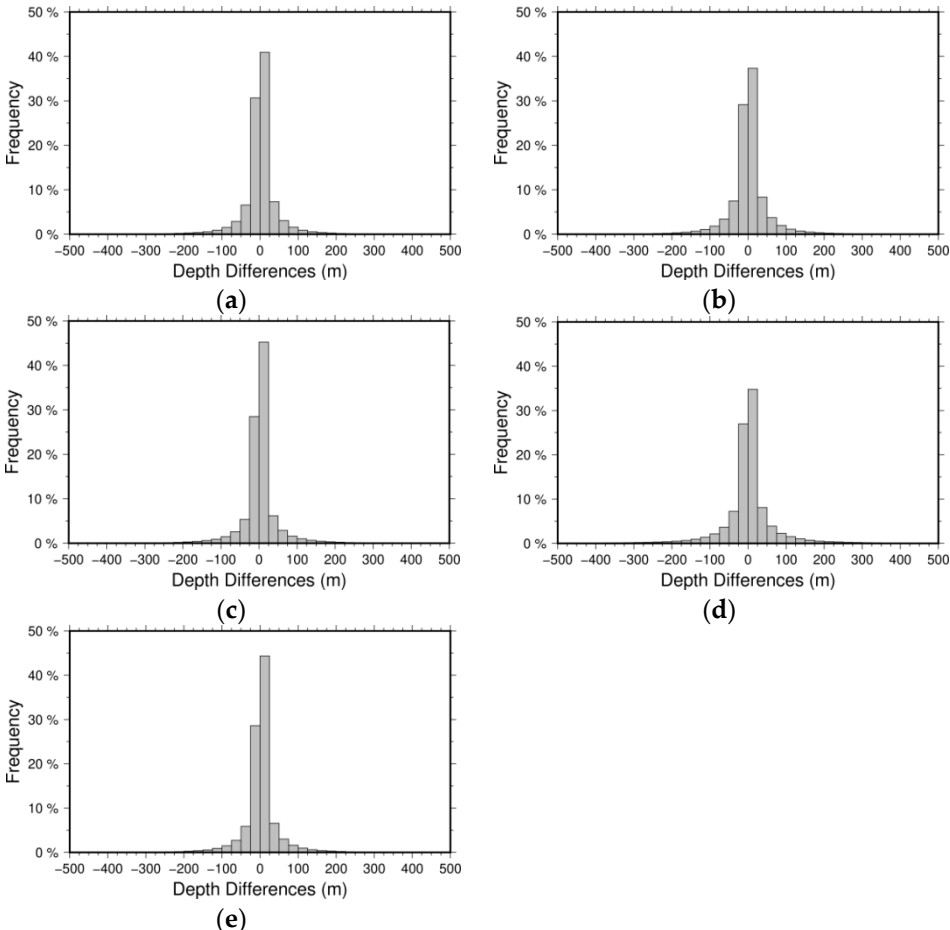

**Figure 12.** The frequency distribution histogram of the differences between BAT_VGG2021 and ship soundings in north Pacific (**a**), south Pacific (**b**), north Atlantic (**c**), south Atlantic (**d**), and Indian Ocean (**e**).

### 4.2. Model Evaluated by Independent Multibeam Grids of MH370 Searching Area

The satellite altimetric data play a pivotal role in filling the blanks between ship tracks. We can discuss the important role of satellite data by comparing with ship-alone grids and independent multibeam depths. For example, in the northeastern Indian Ocean (76°~120°E, −44°~−6°N), the ship soundings cover is sparse except for west coast of Australia, as showed by the black lines in Figure 13. The ship soundings shown by the black lines were used to construct the seafloor topography model with satellite data, noted as BAT_Pre. The colored depths were multibeam grids of the MH370 searching area, and were used to assess the predicted model, BAT_Pre, as independent data.

Using a combination of ship soundings (black lines in Figure 13) and the satellite altimetric VGG model, a 1′ × 1′ seafloor topography model (BAT_Pre, as shown in Figure 14) in the northeastern Indian Ocean was constructed using the data processing procedure proposed in Section 3.2 of this paper. For comparison, a ship-alone grid (BAT_Ship, as shown in Figure 15) was also constructed using GMT tools [31]. Comparing BAT_Pre (Figure 14) to BAT_Ship (Figure 15), the VGG and ship predicted model reveals more seafloor details. To the east of the Ninetyeast Ridge, the BAT_Pre model reveals more seamounts around the Raiit Rise, and other ridges and troughs south of the Wharton Basin. Along the southeast Indian Ridge, the BAT_Pre model recovers many troughs relating to transform faults.

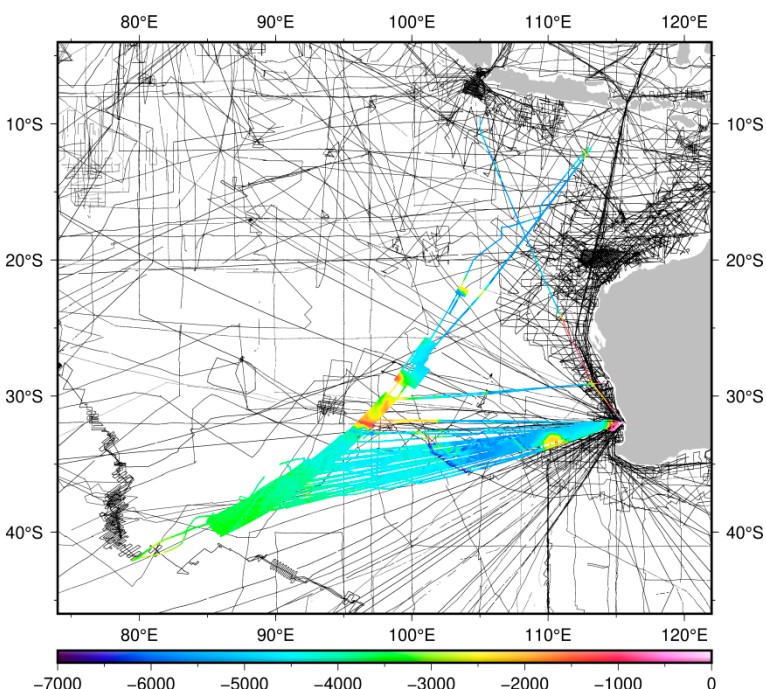

**Figure 13.** Distribution of ship soundings in the northeastern Indian Ocean. The black lines indicate ship soundings used to construct seafloor model with satellite data, noted as BAT_Pre. The colored depths were multibeam grids of MH370 searching area, and were used to assess the predicted model as independent data.

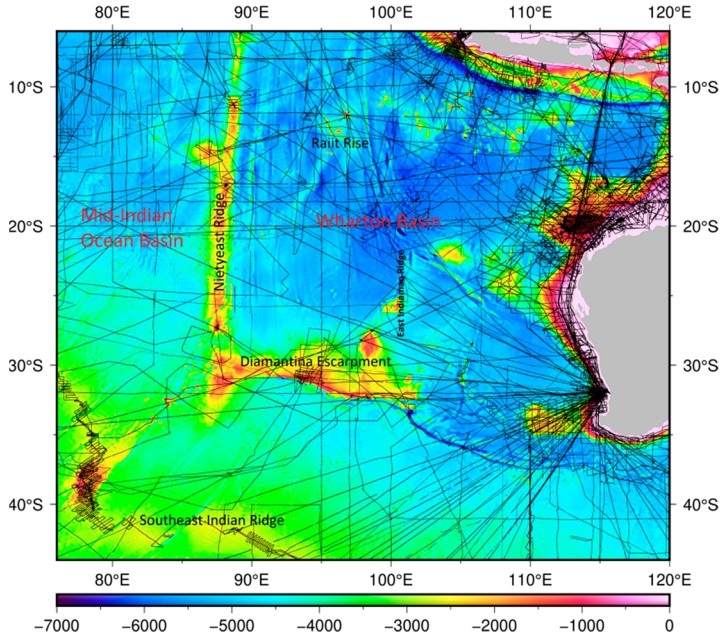

**Figure 14.** A $1' \times 1'$ seafloor topography model (BAT_Pre) predicted from sparse ship soundings and satellite VGG model. The thinblack lines indicate sparsely distributed ship depths used to construct BAT_Pre model. The model reveals seamounts' details and "troughs" along southeast Indian Ridge very well.

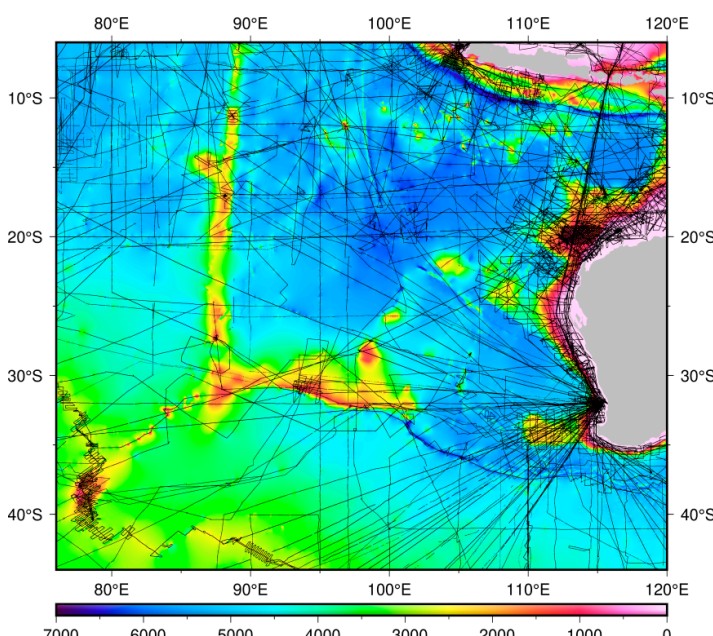

**Figure 15.** A $1' \times 1'$ seafloor topography model gridded from ship-alone data using GMT tools. The thinblack lines indicate all ship soundings used to build BAT_Ship model. The model shows seafloor topography of large scales but loses many details.

The multibeam grids of the MH370 searching area (Figure 13) were not used in the BAT_Pre model (Figure 14) and BAT_Ship model (Figure 15). Thus, we used these grids to assess model accuracy quantitatively. Table 4 gives the differences between the BAT_Pre model, BAT_Ship model, and multibeam grids in the MH370 searching area. The results show that the STD difference between the BAT_Pre and multibeam grids is 106.2 m, better than BAT_Ship which is 148.5 m. Figure 16 shows the differences between the BAT_Pre model and multibeam grids. Figure 17 shows the differences between the BAT_Ship model and multibeam grids. Comparing these two figures, we found that the BAT_Pre model is better, especially around the Diamantina Fracture Zone and East Indiaman Ridge where seafloor topography changes rapidly.

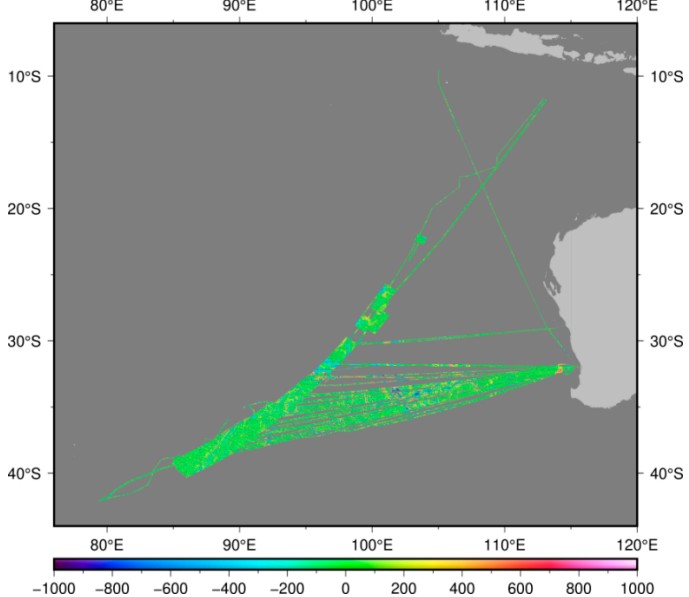

**Figure 16.** The differences between BAT_Pre model and multibeam grids in the MH370 searching area.

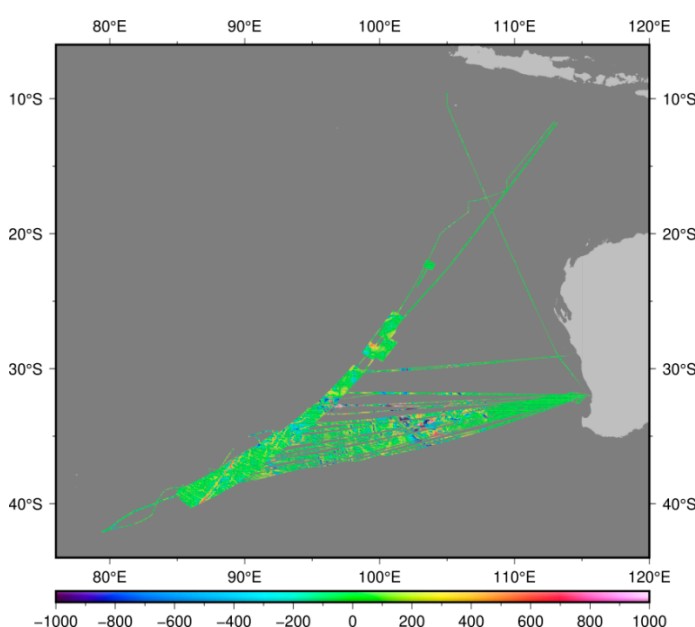

**Figure 17.** The differences between BAT_Ship model and multibeam grids in the MH370 searching area.

**Table 4.** The statistics of differences between BAT_Pre model, BAT_Ship model, and multibeam grids in MH370 searching area.

| Model | Minimums (m) | Maximums (m) | Mean (m) | STD (m) |
|---|---|---|---|---|
| BAT_Pre | −416.1 | 416.1 | 1.9 | 106.2 |
| BAT_Ship | −639.4 | 639.4 | 3.6 | 148.5 |

## 5. Conclusions

We constructed a new $1' \times 1'$ global seafloor topography model, BAT_VGG2021, from ship soundings and satellite altimetric VGG model (SIO curv_30.1.nc). The SIO topo_20.1.nc model was used to constrain seafloor topography at wavelengths longer than 160 km. The VGG model was used to constrain seafloor topography at 15~160 km wavelength bands. The ship soundings were used to calibrate the topography-to-VGG ratios at 15~160 km wavelength bands. The accuracy of the new model was assessed by comparing it with ship soundings and existing models. The results show that the STD differences between BAT_VGG2021 and ship soundings are about 40~80 m. The north hemisphere (north Pacific, north Atlantic) shows higher accuracy than the south hemisphere (south Pacific, south Atlantic), mostly due to more coverage of ship soundings. The new model has an accuracy similar to the SIO topo_20.1.nc model, better than the ETOPO1, DTU18BAT, and GEBCO_08 models, and improved significantly on our last BAT_VGG2014 model, especially in the northern hemisphere. These improvements may be due to the application of the new VGG model and the collection of more ship soundings. The distinct role of the satellite VGG model in seafloor topography modeling was discussed by comparison with a ship-alone grid and independent multibeam grids. The predicted model can reveal more details of seafloor features.

**Author Contributions:** Conceptualization, M.H. and J.L.; ship soundings editing and analysis, L.L.; writing—original draft preparation, M.H.; methodology, M.H. and T.J.; software, M.H.; investigation, H.W. and W.J. All authors have read and agreed to the published version of the manuscript.

**Funding:** This work was supported financially by the National Key Research and Development Project of China (2017YFC1500403-03) and Nature Science Foundation of China (41504017).

**Data Availability Statement:** The model produced in this article may be provided by the author (M. Hu, mzhhu@whu.edu.cn), but it is prohibited to be used for commercial purposes.

**Acknowledgments:** We acknowledge the Scripps Institute of Oceanography (SIO) for providing the latest version of seafloor topography model (topo_20.1.nc) and altimetric vertical gravity gradient anomaly model (curv_30.1.nc). The National Centers for Environmental Information (NCEI) is acknowledged for providing ~74.66 million single-beam depths and ~6.6 GB of multibeam grids. The Japan Agency for Marine-Earth Science and Technology is acknowledged for providing ~120 GB of multibeam grids. Geosciences Australia is acknowledged for providing ~54 GB of multibeam grids. The GMT6.1.1 was used to manipulate some datasets and produce figures in this paper [31].

**Conflicts of Interest:** The authors declare that they have no known competing financial interests or personal relationships that could have appeared to influence the work reported in this paper.

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
