# Peer review of "A New 1′ × 1′ Global Seafloor Topography Model Predicted from Satellite Altimetric Vertical Gravity Gradient Anomaly and Ship Soundings BAT_VGG2021"

_remotesensing, doi:10.3390/rs13173515_

Round 1
Reviewer 1 Report
This study constructed a new 1′ × 1′ global seafloor topography model, BAT_VGG2021, using the satellite altimetric vertical gravity gradient anomaly model (VGG), SIO curv_30.1.nc, and ship soundings. The new BAT_VGG2021 model shows improved accuracy compared to other global seafloor topography models such as DTU18BAT, ETOPO1, GEBCO_08 model, and the last model, BAT_VGG2014. Overall, I found this research very interesting and well-written. I suggest the paper to be published after minor revision. A few minor corrections are as follow:
Page 8, Figure 7, please add (d) at the end of the figure’s caption.
Page 10, Figure 11 has been discussed in the paper. Please refer to the figure in the text.
Reviewer 2 Report
The presented article refers to articles in which a large amount of routine work was performed. It is of more practical than scientific interest. There is one significant remark. Ship measurements have errors. The authors of the article compare the results of their model with ship measurements. As if the ship's measurements have absolute accuracy. Which is completely wrong. As for the model, satellite measurements, ship measurements, it is necessary to cite measurement errors everywhere. Comparing the results of your model with ship measurements, it is necessary to give formulas for calculating errors, absolute, relative, including ship errors.
Lines 169-170. It is said that with the help of ship measurements, data of different quality were obtained. Please indicate on which instruments the measurements were carried out on these vessels. What accuracy does each device give? What are the errors in the measurement data?
Lines 174-175. It is said that in each segment 2° =2°, the depths of the vessel were removed with a difference between the model of the vessel more than twice as large. Please explain why. Because you think that the ship's measurement data is erroneous? Why?
Lines 208-210. The ship's measurements were made with some accuracy. It is necessary to specify this accuracy. And the discrepancy between the model and the ship's measurements should not be better than the specified accuracy of the ship's measurements. The predicted model has an average difference of -0.182 m and a standard deviation of 35.561 m? i.e. an accuracy of about 1 mm? Not happening. Ship measurements give much worse accuracy. Are hydrometeorological conditions taken into account during ship measurements? Surging phenomena, seasonal variations of the level, etc.? What are the ship's measurements linked to? Please explain.
Lines 226-229. The numbers are given: 45,464 m; 100 m; 80,732 m; 200 m. Earlier, I pointed out that the difference cannot be given with an accuracy better than the accuracy of ship measurements. Extra digits should be discarded, rounding the values obtained. These numbers must be brought into line with the errors of the ship's measurements. In addition, when comparing with ship measurements, it is necessary to calculate the elementary errors of the model.
Lines 264-265. Again about accuracy. Ship's measurements give an accuracy of 10 cm? You give a difference equal to 39.6 m or 77.6 m. It follows from these numbers that the ship's measurements were carried out with an accuracy of 10 cm, and the error of which is 5 cm. Throughout the article, it is necessary to give all the calculations in accordance with the errors of ship measurements, if you take them as a standard. But we must take into account that the ship's measurements were carried out on different equipment, each of which has its own errors.
A curious question. About 5000 stationary buoy stations are currently operating in the World Ocean in a continuous mode. Various measurements, including hydrological and hydroacoustic, are carried out at these stations. A huge number of floating and diving buoys, gliders, etc. receive similar information. Can this data be used to fill in the blank spots in the ship's measurements? What will it give? Maybe I should write about this in the conclusion, indicating a separate direction for clarifying my model?
Round 2
Reviewer 2 Report
The authors of the article write that they are building a model and comparing the results obtained with ship measurements. They are not interested in how, on what devices and with what error the ship's data was obtained. A very strange approach. And if the ship's measurements give absolutely unnecessary results? I don't understand the meaning of such comparisons. You can also attach to the "pillar". The whole value of the results obtained is only in the fact that the constructed model gives better results when compared with some "standard" than other models. If other reviewers believe that the article should be accepted, then I will refrain from negative comments and take a neutral position. But I don't understand the scientific value of this article.
Author Response
Please see the attachment.

This manuscript is a resubmission of an earlier submission. The following is a list of the peer review reports and author responses from that submission.